# Data Fusion for Cross-Domain Real-Time Object Detection on the Edge

**DOI:** 10.3390/s23136138

**Published:** 2023-07-04

**Authors:** Mykyta Kovalenko, David Przewozny, Peter Eisert, Sebastian Bosse, Paul Chojecki

**Affiliations:** Fraunhofer Heinrich Hertz Institute, 10587 Berlin, Germany; david.przewozny@hhi.fraunhofer.de (D.P.); peter.eisert@hhi.fraunhofer.de (P.E.); sebastian.bosse@hhi.fraunhofer.de (S.B.); paul.chojecki@hhi.fraunhofer.de (P.C.)

**Keywords:** object detection, edge computing, human-computer interaction, visual analysis, optimization

## Abstract

We investigate an edge-computing scenario for robot control, where two similar neural networks are running on one computational node. We test the feasibility of using a single object-detection model (YOLOv5) with the benefit of reduced computational resources against the potentially more accurate independent and specialized models. Our results show that using one single convolutional neural network (for object detection and hand-gesture classification) instead of two separate ones can reduce resource usage by almost 50%. For many classes, we observed an increase in accuracy when using the model trained with more labels. For small datasets (a few hundred instances per label), we found that it is advisable to add labels with many instances from another dataset to increase detection accuracy.

## 1. Introduction

Machine vision is an important part of industrial production. In particular, collaborative robots, which find increased usage in industrial applications, need to recognize objects and human presence, as well as hand-gesture commands to enable safe human–robot interaction [1]. Such robotic systems often support automated visual quality-control processes to overcome the disadvantages of manual visual inspection, i.e., subjective error assessment and inspector fatigue [2].

Due to such challenges, various methods employing artificial intelligence (AI) have been widely studied to enable automated optical inspection systems. Recent advancements in deep learning have enabled faster and more automated training of networks and achieved reliable detection results. Therefore, deep-learning models are becoming powerful tools for defect detection [3,4] and other machine vision tasks [5]. However, real-time inference with these models requires a significant amount of computational power. The installation of sensitive hardware directly next to the manufacturing is not always possible, nor is it advisable.

Datacenter- and cloud-based computing have traditionally provided an advantage over dispersed, localized hardware [6]. They are relatively inexpensive and massive amounts of information can be processed on demand. However, a major disadvantage is that the data must be transmitted to very distant centralized locations for processing. The results must then be sent back for display or decision-making. This back-and-forth communication adds latency and reduces the speed and interactivity at which an application can run, especially over channels of limited bandwidth. Edge-computing processes data close to where it was generated, and is used in enterprise applications to shorten communication delays. When moving from local computing to edge computing, more than just the processing location should change. When previously different functionalities were distributed over several specialized devices, now similar algorithms can run on the same edge device. In our use case, very similar object and hand-detection systems are executed on the same machine. As pointed out by Padmanabhan et al. [7], GPU memory is a significant bottleneck when running many Convolutional Neural Networks (CNN) in parallel.

In this paper, we test different ways to avoid the bottleneck of GPU memory when running several object-detection tasks in parallel. We focus on an edge-computing scenario, where two similar neural networks are running on one computational node. We compare the performance and the accuracy of domain-specific models, trained to recognize independent object or hand-gesture classes, versus one model that is trained to recognize both categories of classes simultaneously. We test the feasibility of using a single model with the potential added benefit of reduced computational resources against the potentially more accurate independent and specialized models. In contrast to [7,8], we do not modify the CNN networks themselves, but optimize the training set only.

Our research is motivated by the optimization process in a complex edge-computing scenario. In such setups, there are numerous aspects to consider. Although we only focus on one aspect of object-detection optimization, in the related-work section and in Section 3 we also talk about where some side conditions come from and discuss further potential for improvement.

Our results show that using one single model instead of two separate ones can reduce resource usage by almost 50% without decreasing the performance with respect to processing frequency and accuracy. Based on these results, we investigate the optimal split of the batch size and the number of concurrently running CNNs regarding latency, energy consumption and accuracy. To extend our findings, we tested the approach of merging datasets on different splits of the COCO [9] benchmark dataset using multiple object detection networks. These tests confirmed the results from our own dataset.

## 2. Related Work

In our scenario, several components work together. First, we give a very short introduction related to our components. After that, we give a deeper overview of the image-processing queue optimization in the context of edge computing.

### 2.1. Robot Control in Edge or Cloud

The emerging trend of cloud computing allows moving computers that control robots or other actuators in industrial areas further and further away from these sites. Guide et al. [6] clearly expressed that a fog server (a term equivalent to the *edge server*) is a suitable architecture to handle the image-processing workload in human–robot interaction. Several performance metrics of a cloud system are introduced in [10], among them computation time, deadline (latency), energy consumption, and heat generation. However, the authors in Ref. [11] pointed out that the typical objectives in the optimization of edge computing are conflicting, e.g., minimizing the latency can lead to higher energy consumption on the end device, by deciding to execute the tasks locally and vice versa.

### 2.2. Interaction

An overview of the current research direction in human–robot interaction can be found in [12]. More specific to our application is the term human–robot collaboration, where an industrial robot and a human share the same workspace. Although many papers only focus on interaction modality, others investigate when to use which modality [13]. In the context of visual analysis, hand interaction is an important part of our system. Incorporating gesture recognition in human–robot collaboration enhances natural and precise communication, facilitating efficient and effective collaboration in industrial settings. Mohamed et al. [14] give a broad overview of the current state of hand-gesture recognition. In simple cases, hand-gesture recognition can be done using a CNN-based object detector [15].

### 2.3. Inspection

Neural networks are actively being used in industry and manufacturing for quality control and quality assessment. An excellent evaluation of the existing methods and the training datasets leads to the conclusion that many existing models have been trained on highly synthetic datasets and more data based on specific real-world applications is needed [16].

In [3], the authors used a Capsule Network to accurately detect defects on steel surfaces. The network was trained to use the features extracted by another network, which in turn was pretrained on the ImageNet dataset. In another line of research [17], a Scaled-YOLOv4-based object-detection model was proposed, achieving a satisfactory precision level and processing speed on modern consumer hardware. Their model proved the feasibility of using CNNs for reliable real-time defect detection in the industry.

### 2.4. Object Detection

In our application, the tasks of gesture interaction and visual inspection can be reduced to object detection. In a comprehensive review [18], it is stated that one-stage detectors such as the YOLO family perform faster than two-stage frameworks. However, this speed superiority comes at the price of performance degradation for detecting small objects. In our scenario with user interaction in an inspection task, detection should be as fast as possible to minimize latency for the user and machinery, without reducing detection accuracy.

There are only a few works studying the simultaneous use of multiple networks. Lee et al. showed in [19] that it can be better to use three networks to detect holes, sand, or dent defects in a workpiece than a single one that detects all defects at once. In contrast, another study [15] showed that a single network for hand detection and hand-gesture classification can outperform a two-stage design.

### 2.5. Multi-Task CNN

For different tasks, very similar CNN are used. In [20], Foggia et al. tested different strategies to recognize gender, age, ethnicity, and emotion with a single CNN. They compared a single proposed network against a single network with multiple task-specific classification layers, a single network with multiple task-specific convolution and classification layers, and a single network with multiple task-specific convolution and classification layers with an additional feature-sharing layer. In all metrics (accuracy, speed, and memory consumption), the variants with a common base layer performed better, e.g., memory consumption was up to four times lower.

### 2.6. Model Pruning

In model pruning, redundant network parameters are removed to shrink the size of a CNN. Zarski et al. [21] combine transfer learning and pruning 95.4% parameter of a VGG16 model while the accuracy is increased, and computation time drops to 17%. However, in another research, it is reported that for a YOLO5 model, there is no speedup due to pruning, but up to a pruning of 30% of the model parameters the accuracy is nearly the same [22].

### 2.7. Optimizing Image-Processing Queues in Edge Clouds

There are numerous goals for optimizing (performance metrics) an image-processing queue in an edge cloud, e.g., latency, power consumption on (mobile) devices, power consumption overall, or scalability. An overview can be found in [23].

An overview of diverse ways of image-processing queue optimization is given in [24], and [25] investigates latency–energy optimization in a mobile AR application.

For an edge server with multiple models (object detection and grasp planner), it is demonstrated [26] that increasing batch size increases the latency. In [27], it is shown that despite an increase in latency the throughput of an edge system can be increased by increasing the batch size, providing better load balancing. The influence of different factors on power consumption in an edge system is evaluated in [28]. They emphasize scheduling strategies and non-GPU factors as important factors. For an image classification task with Resnet-50 and a given latency of 100 ms, the researchers in [29] measured a maximal batch size of 26, while the GPU utilization stays below 28%. In this configuration, the potential of the GPU is only partially used.

GPU memory limits the number of concurrently running models and the batch size. Srivastava [30] found that batch size vs. memory usage not only depends on the model but also on the specific GPU version (in this case, Nvidia P100 vs. Nvidia V100). The results from [31] show the potential of optimizations e.g., using float16 instead of float32, but also that batch size vs. latency tradeoff is similar for different models. Therefore, it seems that measurements on a not fully optimized processing queue (e.g., using PyTorch model instead of ONNX models) are possible. Principle guidelines for the optimization can be found, but surprising changes in exact parameters should be expected even when changing the GPU architecture.

### 2.8. Optimization of CNNs for Cloud Computing

A research approach similar to our work is described in [8]. Instead of fusing datasets, they fuse trained models to improve inference time on a server running multiple models. For running two ResNet-50 vs. one fused model, they obtain a speedup of 2.6. Although in concurrent mode it is impossible to run 16 models in parallel, the fused network can handle up to 32 models at the same time. The researchers in [32] extend this approach to more diverse models and discuss many challenges of this approach. They bring out the huge saving of memory and training costs, but they did not mention accuracy and latency topics. In [7], the work is continued, and it is shown that memory consumption can be reduced by up to 60.7% while accuracy is increased.

The low utilization of the GPUs is addressed in [33]. To increase it, the parallel execution of multiple kernels is proposed. In contrast to [7,8], only the scheduling of the different kernels is modified, not the models or weights of models.

Rivas et al. [34] uses the edge server to train specialized models, which can run on the camera. They aim at creating small device-specific networks that outperform the generalized network. Similarly, the authors in [35] compared a YOLOv7 model [36] trained on MS COCO data set to YOLOv7 tiny models trained on 18, 4, and 6 classes, respectively. It was found that the small models have a comparable accuracy, but run more than 300% faster.

Another approach to overcome the restriction of the number of parallel running models is to process only “active regions” (regions with motion). Multiple active regions from different streams can feed into one model and significantly increase the processing rate [37].

In summary, it would be advisable to run only a few models on an edge server. If a latency restriction applies, such as in real-time processing tasks, the maximal batch size is usually lower than the theoretical maximum allowed by the hardware. Therefore, if using an edge server with multiple detection tasks, it might be beneficial or required to handle different tasks simultaneously within a single model. The authors in several other papers pointed out how important reducing the numbers of models on a server can be, but it remains to be evaluated if there are other promising approaches to do this, and also collect more measurements to investigate the improvements [8,32].

When adapting models, it would be good to know how many classes the CNN can handle without losing accuracy. In [38] a current overview of the theoretical modelling of the model capacity of a CNN is given. They noted that the effective complexity of a trained deep model may be much lower than the expressive capacity. But today, no approximation of expressive or effective capacity can be provided. Since the effective capacity of a model is highly influenced by training data, we think it is a reasonable strategy to modify training data (e.g., merge them) instead of modifying the network.

## 3. System Architecture

In our scenario, we perform an interactive, robot-supported visual inspection of a rectangular metal construction piece that can be placed on a conveyor belt to be transported between the user and the robot (see Figure 1).

The entire visual inspection pipeline can be controlled via simple hand gestures. The human operator can indicate to the system which construction pieces need to be remembered and processed, as well as start and stop processing. The system can automatically stop the robot from moving if the operator’s hand position comes too close to it. The object-inspection system detects the objects on the conveyor belt and analyses them and the screw/hole layout. The system also verifies the object and its position for processing, as well as remembering the objects indicated by the operator. System states and messages are displayed on a screen using a web-based interface.

Our system is a test bed for edge-based human–robot interaction, as shown in Figure 2. The production site setup consists of a Kuka robot, conveyor belts, and GigE-Vision Basler acA1920-40gc cameras (Basler AG, Ahrensburg, Germany), while the edge server with one Nvidia A100 GPU (Nvidia, Santa Clara, CA, USA) is located at our edge site 2 km away.

Our network architecture is similar to [39], with the difference that our design is focused on more centralized processing to reduce the number of computation nodes (one goal of cloud-like computing). We also virtualized the Programmable Logic Controller (PLC) and moved it to our server. That way every component is executed on the server. Edge server and production sites are connected by one private 1 Gbps Ethernet connection. To separate robot control and camera data streams on the same connection, all traffic was managed via Time-Sensitive Networking (TSN). The time-synchronization, provided by the TSN, ensures the clear separation between the video streams and data streams used for the robot control. A total of 100 Mbps of TSN traffic was reserved for robot control and 400 Mbps for the camera connection. On the edge server, TSN runs as a software module. At the production site, TSN-connectors are the gatekeeper between the TSN system and the non-TSN camera- and robot hardware. The overall ping time from the edge server to the cameras was below 2 ms.

As suggested in [40] an OPC UA server is used to control the robot and the conveyor belts. The OPC UA server and a virtual PLC are both running on the edge server which is used for image processing.

To achieve the lowest latency and highest image quality, uncompressed raw video streams (Bayer pattern) are transmitted. Using the multicast feature of the GigE-Vision camera, the images can be sent to two separate applications at once.

## 4. Experimental Setup

In this section, we provide an in-depth exploration of the datasets and models employed in this study. Here we give a detailed description of the datasets developed for training the model while examining and comparing various object-detection models to identify the optimal choice for subsequent training.

### 4.1. Datasets

The following section describes the datasets used for training the model, ensuring a comprehensive understanding of their composition and characteristics.

#### 4.1.1. Our Datasets

Datasets for training and testing consist of color images with a resolution of 1920 × 1200 pixels containing hand gestures and/or the construction piece, which is diagonally symmetrical, contains straight horizontal and vertical grooves, and has six identical screw holes, as depicted in Figure 3. The training dataset used by our visual inspection system included around 26,000 images in total. For the purpose of this research, we have extracted a balanced subset from it consisting of roughly 6000 images. For the training validation, we randomly sample 10% of our training subset. The test dataset contains approximately 5200 images in total.

In the training set, at least 64% of the images contain the object, with on average 3 screws or bolts inserted into the holes in different combinations. At least 67% of the images contain one or two hand gestures. The more natural neutral hand position, represented with the “other hand” label, is found in 40% of the images. The remaining four gestures are distributed evenly throughout the dataset.

In the test set, 72% of images contain the object and 92% of images contain hand gestures, with a more even class distribution of the latter.

The bounding boxes around the construction piece range from 1% up to 7% of the total image area, while for the screws and screw holes, the sizes of the bounding boxes take up from 0.01% to 0.3% of the total image size. Finally, depending on the gesture, the bounding boxes around the hands take up 1% to 24% of the total image area.

#### 4.1.2. Generalized Dataset

To generalize our results, we created three subsets from the commonly used COCO [9] benchmark dataset. Our objective was to examine whether the same effect observed in our own dataset could be replicated. Upon analyzing our initial object and gesture datasets, we observed the presence of contrasting properties:different knowledge domains, when merging a dataset containing metallic construction pieces with screws and bots with a dataset containing human hand gestures;vastly different object sizes, with screw holes taking up less than 0.3% of the total image area, and hand gestures—up to 24%;unbalanced number of instances per label, where every construction piece always has six times as many screws/holes, the “other gesture” class dominates the hand dataset, and the small number of “rejected part” instances

To investigate all three cases, we extracted several groups of subsets from the COCO dataset, which have similar properties:*Domains* split, containing three subsets “Animals”, “Food” and “Sport”, with 10 classes in each, with a similar distribution of the number of labeled instances*Number of Instances per Labels* split, containing two subsets—one with the top 10 classes that have the highest number of labeled instances, and another with the 10 classes that have the lowest number*Box Size* split, with two subsets, containing the top 10 classes with the largest average object bounding box sizes, and 10 classes with the lowest average box sizes

Statistical data on the number of instances per class and box size can be found in the result section.

Additionally, we investigate the effect of adding the top 10 most labeled classes to other subsets, as well as our own object-hand dataset.

### 4.2. Models

We compared various popular object-detection models such as SSD (Single Shot MultiBox Detector) [41], YOLOv5 (You Only Look Once) [42] and Faster-RCNN (Faster Region-Based CNN) [43], as well as different model sizes of YOLOv5 and SSD.

The model chosen for our visual analysis system was YOLOv5 [42], which belongs to a family of object-detection architectures and models. The network architecture of YOLO is divided into three parts: a backbone, a neck, and a head. The backbone is a series of convolutional layers that extract features from the input image. The neck combines features from different scales to improve object detection. The head is a set of fully connected layers that predict the location and class of each object in the image.

YOLO divides the input image into an *S* × *S* grid. If the center of an object falls into a grid cell, that grid cell is responsible for detecting that object. Each grid cell predicts B bounding boxes and confidence scores for those boxes. During inference, the resulting boxes are filtered by their conference and reduced using non-maximum suppression.

YOLO models solve object detection as a regression problem and provide the class probabilities of the detected images, requiring only a single forward propagation through a neural network to detect objects in real time.

There are different variants of YOLOv5 model (Small, Medium, Large and eXtra large). Models differ in their depth and layer channel width, and resulting from this the performance in terms of latency and accuracy is also different. For the SSD models, we used the VGG backend and two options for the input size: 320 and 640. For the Faster-RCNN model, we used the ResNet-50 backend with an input size of 512. For our configuration and the workpiece dataset, we found the results given in Table 1.

We have chosen the medium-sized YOLOv5 model (YOLOv5m) for our case, as it provides a good balance of performance and accuracy. Input images for the YOLO detector are scaled to 640 × 448 pixels. The aspect ratio was selected to best fit with the aspect ratio of our cameras.

Similar to [44], we saw that on some labels YOLOv5s performs better than YOLOv5l. However, due to the better average value of mAP@0.5 we settled on using the YOLOv5 model.

We used transfer learning with a model pretrained on the COCO dataset [9], and trained our models using the Adam optimizer, with a starting learning rate of 0.001 and a batch size of 12.

In total three models were trained, each one for 250 epochs: the object-detection model (ODM) for the four workpiece classes (*accepted part*, *rejected part*, *screw hole* and *screw head*), the hand-detection model (HDM) for five gesture classes (*pointing*, *thumb up*, *diver’s OK*, *full hand* and *other*), and a combined detection model (CDM) for all nine classes simultaneously. These classes were specifically selected for our human–robot interaction scenario, described in Section 3.

The ODM and HDM were trained iteratively, by annotating parts of the training dataset, training the models, using them to automatically annotate the next batch of the dataset, manually correcting the annotations, and then training again. For our tests, we used a finalized version of the annotated dataset.

## 5. Results

The following section presents the experimental results of our research. At first we present the accuracy evaluation for the different splits from our own dataset and from the splits from the COCO dataset. In the second part we analyze runtime and memory consumption.

In our experiments, we trained 31 different combinations of YOLOv5 networks and subsets from COCO or our own dataset. Our training took 25 to 46 h for the YOLOv5 models, and 6 to 48 h for the SSD model, on a cluster containing V100 and A100 GPUs. Each model was trained for 100 epochs using 4 GPUs in parallel, with a batch size of 16. To eliminate the influence of any other variables, all other training hyperparameters were left at their default values, recommended by the corresponding object-detection models.

### 5.1. Accuracy

In this section, we evaluate the accuracy of our trained models on both our datasets and various splits of the COCO dataset.

#### 5.1.1. Our Own Dataset

The mean average precision for the detection of objects and gestures with different models is shown in Figure 4.

We use our annotated testing dataset to evaluate the accuracy and computational load of the three models ODM, HDM, and CDM. We used the Mean Average Precision (mAP) measurement with the Intersection over Union (IoU) threshold of 0.5 for our testing, henceforth referred to as mAP@0.5, as well as the Mean Average Precision averaged at 10 IoU thresholds in the range from 0.5 to 0.95, henceforth referred to as mAP@[0.5:0.95] [45].

mAP@0.5 provides an overall measure of the model’s ability to accurately detect objects with reasonable localization, while mAP@[0.5:0.95] takes into account both high-overlap and low-overlap detections, offering a broader assessment of the model’s ability to handle varying levels of object localization, and giving a more comprehensive evaluation of the model’s performance across a range of object sizes and shapes [46].

We calculate the mAP@0.5 for the four object classes using ODM and CDM, and do the same for the five hand classes, with HDM and CDM, respectively. Figure 5 shows the precision-recall curves for the aforementioned cases, as well as the mAP@0.5 values, averaged over classes.

The differences for precision, as well as mAP@0.5 and mAP@[0.5:0.95] measurements between the specialized (ODM and HDM) and combined CDM, grouped into triples for every class separately, are shown in Figure 6 and Figure 7.

The figure shows that the overall mAP@0.5 for the object classes as detected by ODM is higher than the mAP@0.5 for the hand classes by the HDM. This can be explained by a higher variety in the appearance, position, rotation, and complexity of the hand classes compared to the more uniform-looking object classes. The mAP@0.5 measurement for only the object classes using the combined CDM was slightly lower compared to the specialized ODM, while the CDM performed slightly higher than the HDM for the hand classes.

#### 5.1.2. Split by Domain

Merging classes from different domains is the most typical case in practice. We use three similar splits here (sport, food, animals), each containing 10 classes with between 1200 and 4500 labels each, and the size variation between 2% and 23% of total image area (see Figure 8, Figure 9 and Figure 10). The validation results for all four models are shown in Figure 11. We can see from Figure 12 that the accuracy of the combined YOLOv5 model for all class categories is comparable to the separate specialized models, which is consistent with our practical results of combining object and hand detection.

#### 5.1.3. Number of Instances per Label

A possible issue for the training of the CNN is an unbalanced dataset. Therefore, we tested an extreme selection from COCO. In Figure 13 and Figure 14 the number of instances per label for the ten most and least frequent classes is shown. The most frequent instances range from 66,808 (people) to 5805 (bench) and the least frequent labels range between 1601 (microwave) and 198 (hair drier) instances. Although this distribution is contrary to training recommendations, it is quite typical for practical scenarios where instances of defects or negative cases, for example, are infrequent.

Although for classes of the most frequent subset, there is no difference in accuracy, the effect for least frequent classes is impressive, see Figure 15 and Figure 16.

#### 5.1.4. Split by Average Box Size

The other selection with possible difficulties was the difference in the box size of the instances. The biggest box size was 40% (bed) to 13% (dog), as shown in Figure 17, and the smallest 0.2% (sports ball) to 1.2% (handbag), as seen in Figure 18.

Figure 19 and Figure 20 show that mixing the classes with small and large bounding boxes gives only a moderate increase in detection accuracy.

#### 5.1.5. Increase Accuracy by Adding Labels with Many Instances

When testing the split by the number of instances, we observed that adding labels with many instances improves the accuracy of the labels with the low number of instances. Therefore, we test if we can reproduce these results when the number of instances is larger than in the split by the number of labels. In our test, we added the split with the most labeled images (see Section 5.1.3 to the “Domains” split (see Section 5.1.2)).

As evident from the results in Figure 21 and Figure 22, adding the annotations from the amply labeled subset to other subsets produces the same expected result: the prediction accuracy on the classes from the tested subsets increases.

To investigate whether this would also benefit our visual analysis models, we merged our object and hand datasets with the same COCO subset of the top 10 most labeled instances. The results can be seen in Figure 23.

#### 5.1.6. Overall

In Figure 24 the composition of all three splits is shown. For all YOLOv5 models trained with the combined data set the performance indicators (precision, recall, mAP@0.5 and mAP@[0.5:0.95]) are better than the specialized ones. For both SSD models, there is only a small difference with slightly decreased performance indicators.

#### 5.1.7. Statistics

For each label, we obtain different results between the separate and combined modes. Therefore, looking at mean values overall may be misleading since the differences can be produced by random factors and are not statistically significant. For a parametric test, the preconditions are not met, because results per label do not show a normal distribution. Similar to Zhao et al. [44] and Liu et al. [47] we select Wilcoxon sign ranking test [48] as the most suitable one. It does not need normal distributions and is robust in the presence of outliers. Statistics are computed using Pingouin [49] version 0.5.2.

For all YOLOv5 variants, we found that a combined model is more accurate than specialized models (mAP@[0.5:0.95]) with p<0.001.

For YOLOv5m the conditions (normal distribution) for paired t-test were met, the 95% confidence interval for difference in mAP@[0.5:0.95] is [−0.11,−0.07], t(69)=−8.48, p<0.001, d=0.577. Therefore, the combined model is significantly more accurate, and the effect size is medium and relevant.

For SSD320, SSD640 Wilcoxon signed-rank test did not find a difference for (mAP@[0.5:0.95]) with p=0.1818 and p=0.2727.

### 5.2. Processing Time, Memory Consumption

The performance measurements were made on our dataset only. There is no difference in the splits from COCO dataset.

Performance measurements using an Nvidia A100 GPU are given in Table 2 for image processing (including preprocessing, inference, non-maximum suppression, and postprocessing), as well as the time spent on frame acquisition and communication with the Web server. The processing frame rate is limited to 16.6 Hz due to the camera frame rate. Table 3 shows the average image-processing time for different GPUs in offline mode, to eliminate any possible camera delay. As seen from the table, for our CDM, the current consumer desktop GPUs perform better. We expect edge GPUs to be advantageous if numerous models are used in parallel, or when batch processing is used.

For ODM and HDM running concurrently, we measured around 40% GPU utilization, 4.4 GB of GPU memory, up to 85W GPU power, and up to 2 CPU cores usage. Running CDM only halves computational load (around 20% GPU usage, 2.3 GB GPU memory, up to 40 W GPU power usage, one CPU core).

We observed a side effect of using a combined model versus using two separate models in terms of their behavior when their respective objects do not appear in the frame (e.g., no hands in the frame for the HDM), and the resulting performance loss from the Non-Maximum Suppression (NMS) algorithm, which is either always called in the case of the combined model, or is sometimes skipped altogether in the case of separated systems. From our testing, depending on the model used and the presence of the detected objects in the frame, as well as the number of classes, the overall processing time per frame could be raised on average by 1.7–3.1 ms due to the NMS algorithm.

In the literature, usually, only one process with one model is evaluated with different batch sizes. However, it is possible to run different models in different processes on the same machine.

With our YOLO v5 models and the Nvidia A100 that has 40 GB of GPU memory, we can process a maximal batch size of 122 with a single model, before reaching 100% GPU memory usage (Figure 25), and we can run up to 16 processes, each with its own model and processing six images at a time, before hitting the GPU memory limit. We run the tests using Python 3.8, PyTorch 1.11 with CUDA 11.1 running on a 16-Core Intel Xeon CPU and an input image size of 640 × 480.

In our test scenario, we perform every preprocessing step, including image scaling, format conversion, and channel transposing, aside from handling incoming images. On every run, we allowed for a warm-up period of 10 s, where the model inference is running, after which we started the measurements. For every case of batch size and process number combination, we ran a total of 500 iterations.

In our use case of robot control, the typical metric of frames per second is not sufficient (Figure 26), because we have an upper limit of reasonable latency. Thanks to batch processing, a huge number of images can be processed at the same time without increasing the total processing time. To use the advantages of batch processing in real-life use cases, many video streams have to be combined, which introduces additional synchronization overhead. As shown in Figure 25 for up to a batch size of 24 the latency stays constant. We did not have an explanation for the drop in memory usage that happens at a batch size of 122. No errors in output or any sudden drops in latency were observed. For our use case, we set a soft limit on the maximal allowed latency of the object detection at approx. 60 ms, similar to the frame time in our initial application.

When operating under the latency limit, our edge server can process up to 68 different video streams with the same model in one process. We tested different combinations of the number of processes and batch sizes and checked what is the maximum number of simultaneously processed images. We found that up to 8 processes with the appropriate batch size around, 48 images can be processed within our latency limit of 60 ms. For a larger number of processes, this number decreases. With 10 different models 40 streams are possible, with 12 different models 36 streams are possible, for 14 models—28 streams, and finally, for 16 models only 1 image per process is possible within a reasonable latency limit.

Figure 27 shows the average latency depending on the number of concurrently processed images. We can see that reducing the number of concurrent processes also decreases the average latency for the same number of concurrently processed images.

We verified our results from the Nvidia A100 with an Nvidia Titan RTX on a server with a Dual Xeon Gold 5128 using Python 3.8 and PyTorch 1.8 with CUDA 10.2. The results are shown in Table 4. The maximum possible batch size for one single model was 19 and on the other hand, it was possible to run up to 10 models in parallel.

When using CUDA 10.2 instead of CUDA 11.1 we obtained an increased latency, especially for a single process and a batch size of 1 (12.7 ms vs. 19.1 ms), which seems to be a known issue at the moment (see Table 4).

## 6. Discussion

In Figure 6 and Figure 7, it is shown that the mAP values for our workpiece detection and hand detection are similar in both training setups. Some classes perform slightly better, others show a marginal decrease. All changes are irrelevant to our application. Our processing is conducted on videos and the postprocessing can handle it during detection for a short period. Therefore, in our use case, we benefit from the increased performance by batching images from several cameras; see Figure 27.

Results in Table 4 reveal issues when comparing measurements from different configurations. Newer software versions, e.g., CUDA 11.1 vs. CUDA 10.2 can be slower than older ones. These measurements can vary for different test cases. We cannot find any problems with our software installation or measurement setup. Therefore, for the fine-tuning of an edge system, measurements should be repeated for the target configuration.

In the training guide for YOLOv5 [50], more than 10,000 instances per class and more than 1500 images per class are recommended. In our test with the subset of the COCO dataset, these recommendations are not fulfilled. However, the used number of instances is typical for many public datasets.

For the YOLOv5 model, we found that models trained with a combined dataset perform better than the specialized models. When having a dataset with only a few hundred instances per class, it seems advisable to add some more classes with many instances just to increase the outcome for the classes of the intended dataset (see Figure 15 and Figure 16). For an edge server setup, this is an important message. Instead of using several models, only a few are better. Reducing the number of needed models makes it possible to process more images on an edge server. Images from different sources can feed into a batch and this batch can process by the CNN at once. Therefore, the image throughput will increase.

By combining subsets, often a smaller YOLO model can be used. For example, for mAP@[0.5:0.95] split smallest/largest for group largest, the combined YOLOv5s model is better than separate YOLOv5m (0.67 vs. 0.65) and YOLOv5m vs. YOLOv5l (0.72 vs. 0.68) (see Figure 19). This is not true in all cases, but when optimizing a system for real-time application, this can be very helpful. This performance can be used in any configuration, just by adding labels from a public dataset to one’s own dataset.

For the subset of the most frequent classes, we saw a decrease in accuracy, but it is marginal (less than 1%) and statistically not significant. This finding indicates the great importance of the number of used instances per class for training as recommended in [50].

The approach given by Rivas et al. [34] is attractive. Small, specialized models outperform generalized ones, and so there is no need to run big models on an edge server. Our results demonstrate the opposite. When using a smaller network, the detection accuracy is lower than on a bigger one, even when the smaller model is trained on fewer classes. By training the smaller model with additional classes, we can improve the detection performance. For smaller models, e.g., YOLOv5m vs. YOLOv5x, our tests show lower latency and the possibility to process more images at once, but YOLOv5x is not always more accurate than YOLOv5m (e.g., see Figure 15 or Figure 19).

The approach by the authors in [37], where only active regions are evaluated, can easily be combined with our solution. Probably, the model modifications done by [7,8] can enhance the processing on an (edge) cloud server even more. Although our technique and [37] can simply be applied by an application programmer, both other methods should be supported by a toolkit.

Our tests were limited by the existing image-detection dataset. We selected our test cases based on the possibilities COCO offers. It would be good to run a test for a variety of numbers of classes, where all classes have the same number of instances and parameters, e.g., size of objects, aspect ratio, and image contrast, are similar. Therefore, the maximal number of classes a network can process at a given accuracy can be found. Furthermore, the complexity of datasets should be investigated. From our application, one question about the complexity of the dataset is the range of orientation of objects within a class.

Although we tested our idea on YOLOv5 and SSD, the results can be different for other object-detection algorithms. The contrary findings by [34] indicate that more combinations of different object-detection networks and different datasets can uncover other suitable configurations in different edge–cloud configurations.

## 7. Future Work

Samples, where different object detectors are needed in a single system, can be found in many fields. For an autonomous driving system, the authors in [51] describe a setup where multiple object detectors process images from multiple sensors. In such scenarios, data fusion should be tested as one simple optimization strategy.

Newer, bigger datasets, e.g., Open Image dataset [52], should be used to figure out the capacity of a specific object-detection network and the optimal features (e.g., number of images, number of classes, number of instances per class) of the dataset for that model. Maybe pruning approaches similar to those presented by the authors in [21] could help to minimize the memory footprint.

In the related-work section, various ways to optimize object-detection algorithms in edge–cloud environments were shown. Several of them can be combined and others are opposed. In the long term, the different optimization methods introduced in the related-work section should be combined.

## 8. Conclusions

We demonstrated, on our own dataset and on a selection of different subsets from the COCO dataset, that increasing the number of classes in a single object-detection model does not reduce the average precision. So, when handling concurrent object-detection models for different domains, a single model for all domains instead of separate models for each domain can be viable. This can be useful to optimize the setup of an edge server.

We have demonstrated that it is possible to save up to 50% of GPU, CPU, and power consumption using one combined object detector for cross-domain objects, instead of several independent ones. This resource-saving is achieved without sacrificing detection performance.

In a synthetic benchmark without non-maximum suppression, we have shown that using a single model instead of 16 separate models can enhance the capacity of a server from 16 up to 68 concurrent video streams for a given latency restriction. In another benchmark, including non-maximum suppression, we showed a significant decrease in latency when increasing batch size and reducing the number of concurrently running models. Different combinations of batch sizes and numbers of processes can be comparable in terms of latency.

In practice, there is still the issue that datasets for different domains might not be available for joint training: object inspection and human–robot interaction may be the domain of different parties, who, according to internal business considerations, or existing external limitations, might want to protect their own data privacy when sharing the knowledge. To diminish the problem with access to datasets, federated learning can be a potential solution. Future work will investigate the use of federated learning [53] to train a combined network without merging datasets.

Compared to [8], where neural networks themselves are merged, our approach is much simpler if full access to the original training data is granted. The result of the increased number of concurrently running models is similar, but unfortunately, the paper does not provide the processing latency, which is important for real-life applications.

From a software architecture point of view, merging several tasks into one singular component might not always be desirable; however, it is advisable to maximize the advantages of cloud setups.

## Figures and Tables

**Figure 1 sensors-23-06138-f001:**
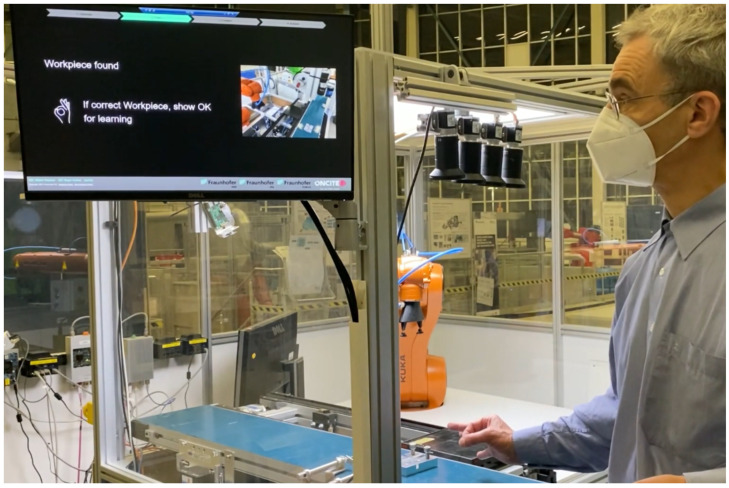
Our actual experimental setup at the production site.

**Figure 2 sensors-23-06138-f002:**
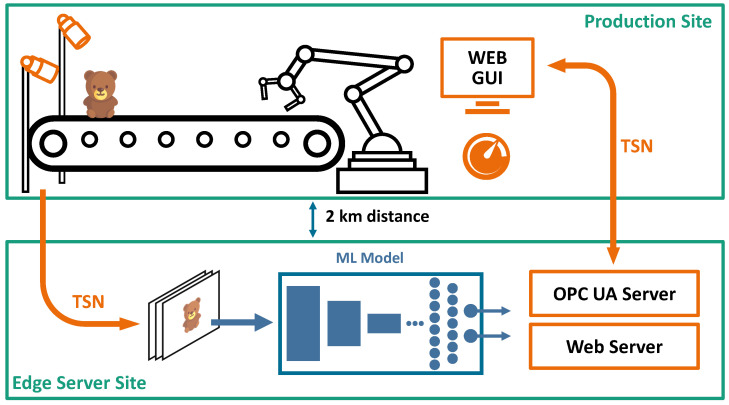
A schematic diagram of the visual inspection pipeline.

**Figure 3 sensors-23-06138-f003:**
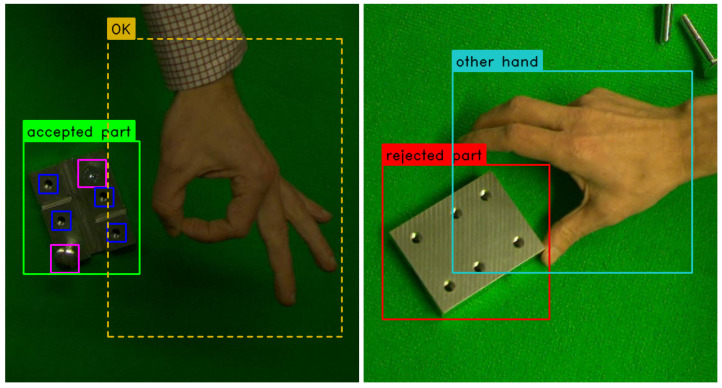
Examples of object classes in the dataset (cropped).

**Figure 4 sensors-23-06138-f004:**
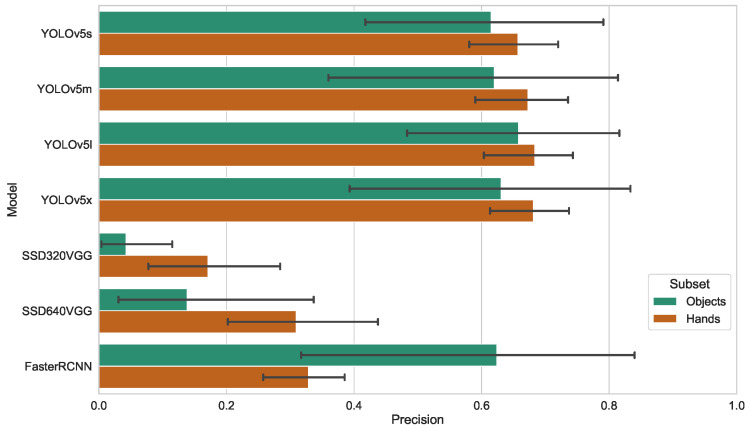
Mean Average Precision metrics for the object and hand-detection models.

**Figure 5 sensors-23-06138-f005:**
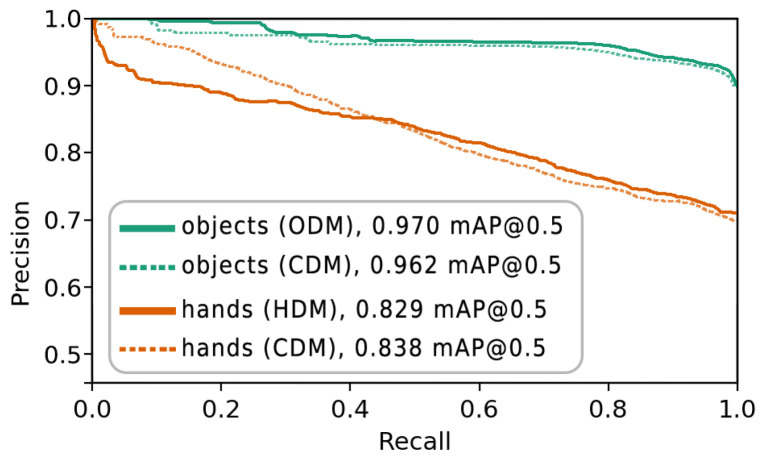
Precision-Recall curve for the independent object detection (ODM) and hand-detection (HDM) models and hands or object detection with the combined (CDM) model.

**Figure 6 sensors-23-06138-f006:**
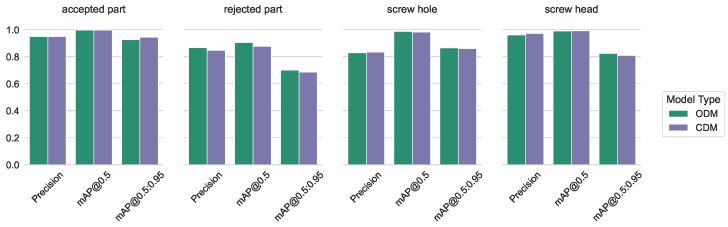
Precision, mAP@0.5, and mAP@[0.5:0.95] measures for the independent and the combined models—for workpiece detection.

**Figure 7 sensors-23-06138-f007:**
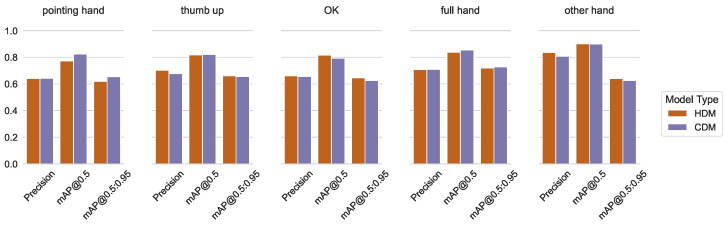
Precision, mAP@0.5, and mAP@[0.5:0.95] measures for the independent and the combined models—for hand detection.

**Figure 8 sensors-23-06138-f008:**
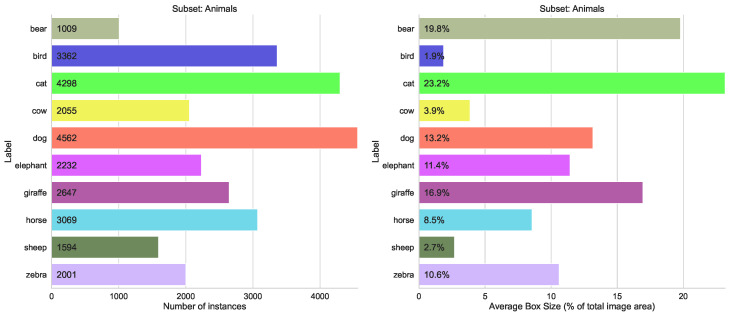
Statistics for the COCO subset with ten classes from “Animals” category.

**Figure 9 sensors-23-06138-f009:**
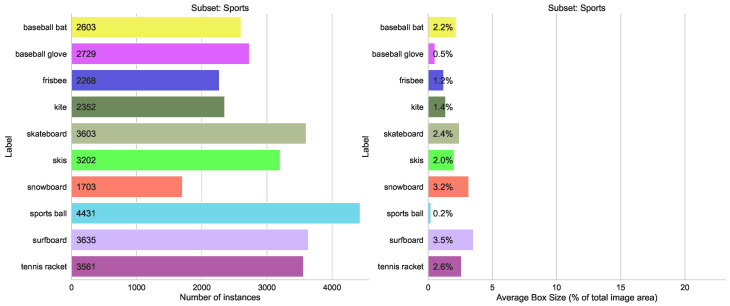
Statistics for the COCO subset with ten classes from “Sports” category.

**Figure 10 sensors-23-06138-f010:**
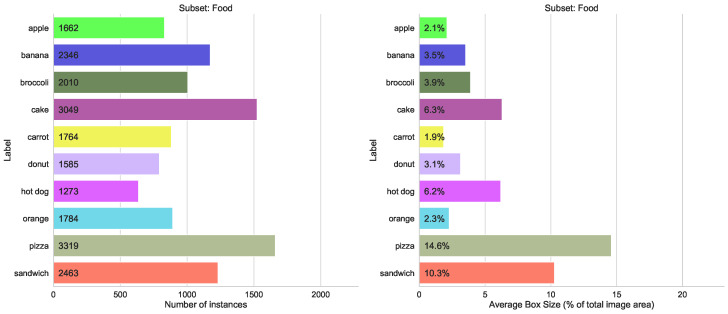
Statistics for the COCO subset with ten classes from “Food” category.

**Figure 11 sensors-23-06138-f011:**
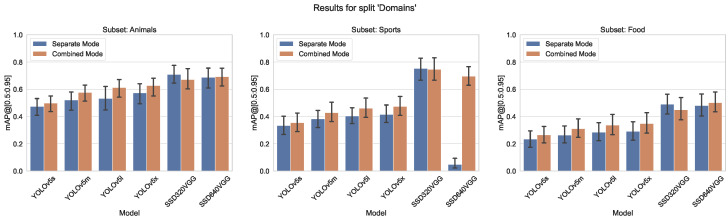
Accuracy metrics (mAP@[0.5:0.95]) of the for different models on split domains.

**Figure 12 sensors-23-06138-f012:**
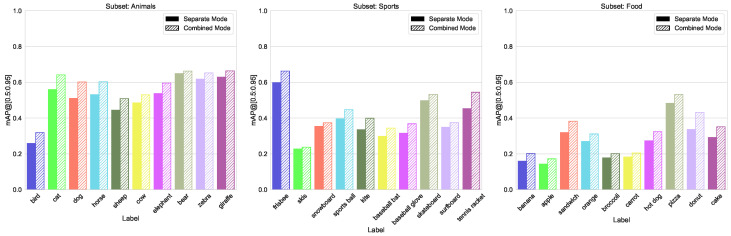
Comparison of mAP@[0.5:0.95] values for both training modes, left group “Animals”, center “Sports”, and right “Food” from split domains.

**Figure 13 sensors-23-06138-f013:**
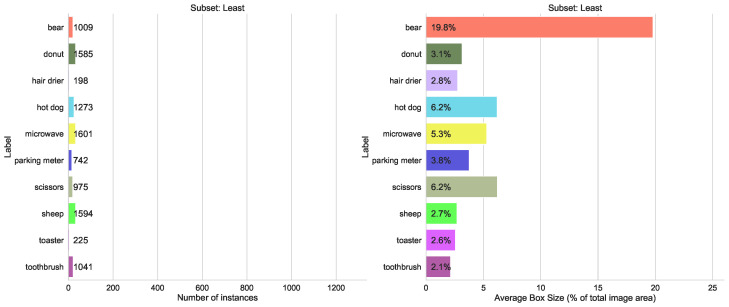
Statistics for the COCO subset with the top 10 least labeled classes.

**Figure 14 sensors-23-06138-f014:**
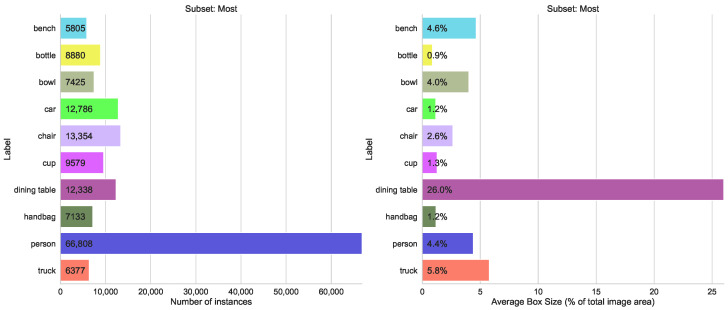
Statistics for the COCO subset with the top 10 most labeled classes.

**Figure 15 sensors-23-06138-f015:**
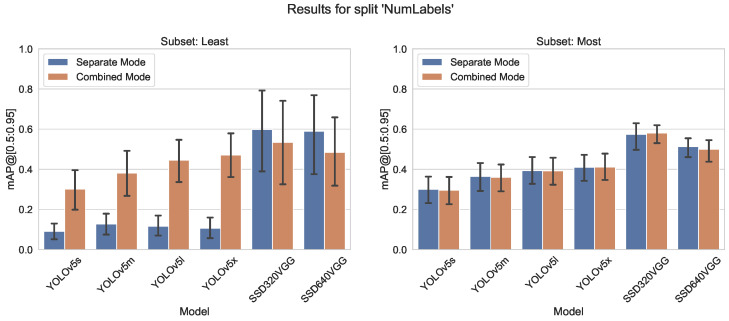
Accuracy metrics mAP@[0.5:0.95] for different models trained on least/most abundantly labeled splits of the COCO dataset.

**Figure 16 sensors-23-06138-f016:**
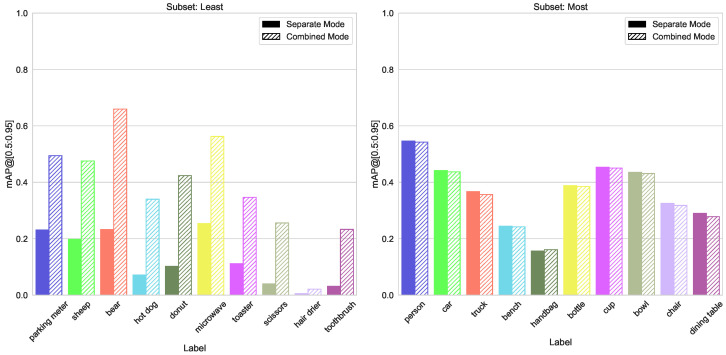
Comparison of mAP@[0.5:0.95] values for both training modes, left group “least”, right “most” from a split number of instances, alternative similar to Figure 20.

**Figure 17 sensors-23-06138-f017:**
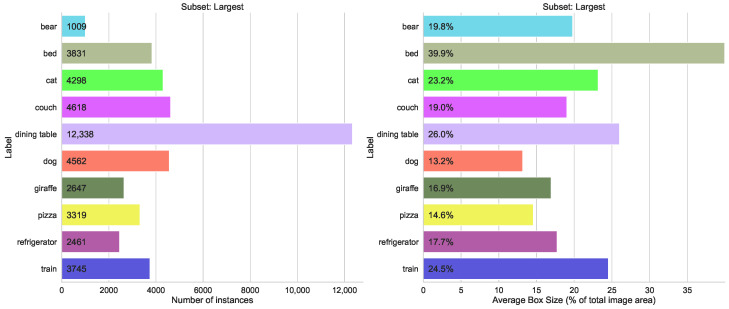
Statistics for the COCO subset of the top ten classes with the largest bounding box sizes.

**Figure 18 sensors-23-06138-f018:**
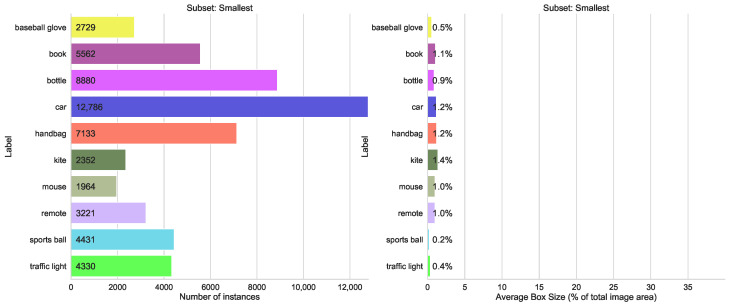
Statistics for the COCO subset of the top ten classes with the smallest bounding box sizes.

**Figure 19 sensors-23-06138-f019:**
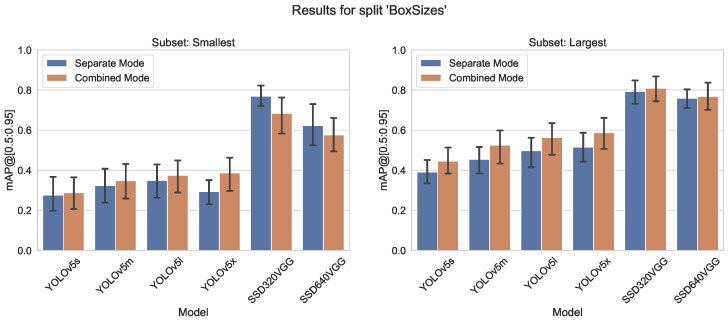
mAP@[0.5:0.95] metrics of the different models trained in separate or combined mode on largest/smallest object box size splits of the COCO dataset.

**Figure 20 sensors-23-06138-f020:**
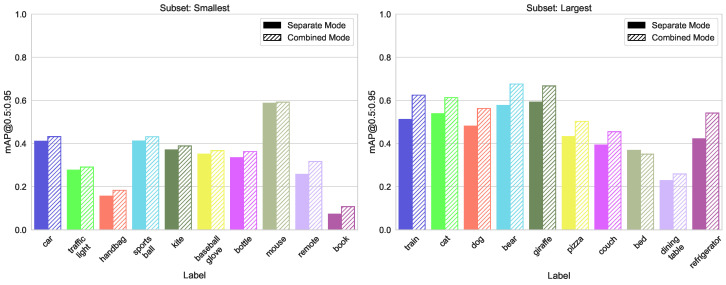
Comparison of mAP@[0.5:0.95] values for both training modes, left group “smallest”, right “largest” from the split “Box size”.

**Figure 21 sensors-23-06138-f021:**
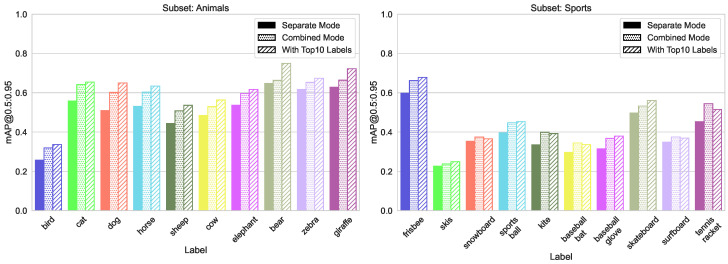
Validation results for models trained on separate and combined subsets “Animals” and “Sports” versus adding the top 10 most labeled classes.

**Figure 22 sensors-23-06138-f022:**
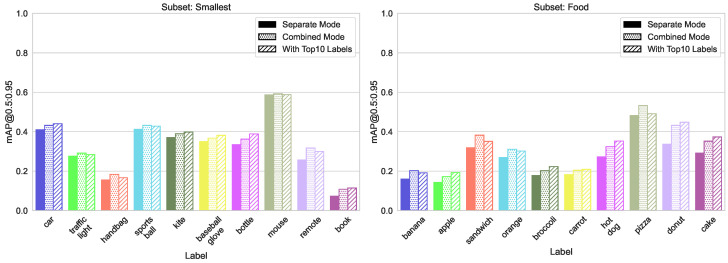
Validation results for models trained on separate and combined subsets “Food” and “Smallest Boxes” versus adding the top 10 most labeled classes.

**Figure 23 sensors-23-06138-f023:**
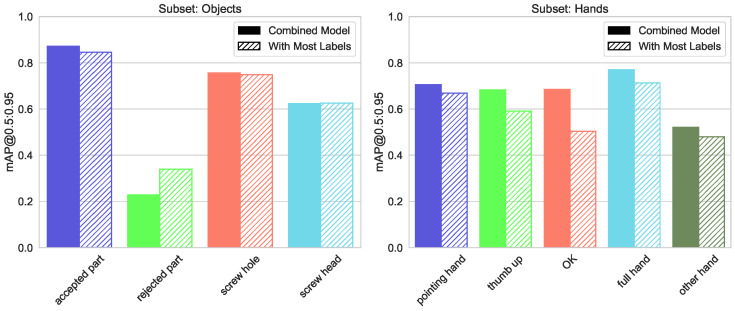
Validation results for CDM model trained together with the data from “Top 10 Most Labeled” subset.

**Figure 24 sensors-23-06138-f024:**
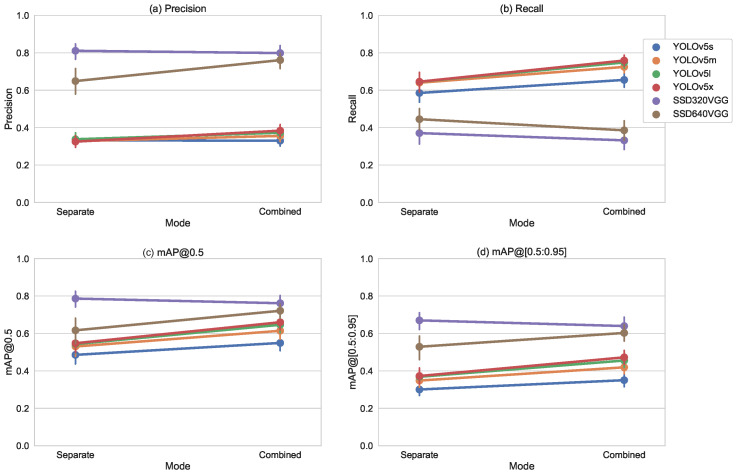
Precision (**a**), Recall (**b**), mAP@0.5 (**c**) and mAP@[0.5:0.95] (**d**) over all three splits, for separate and combined training sets.

**Figure 25 sensors-23-06138-f025:**
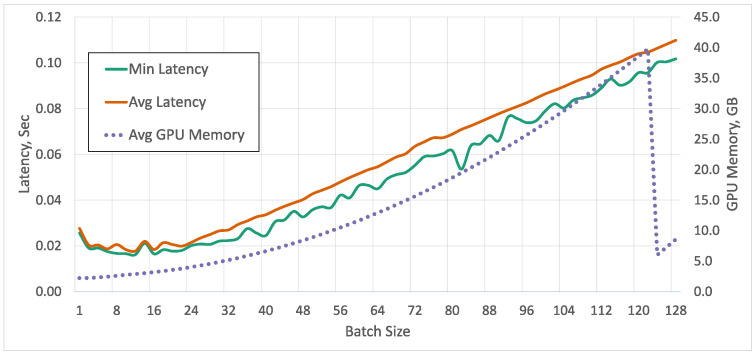
Average and minimum latency of processing different batch sizes with a single model, versus the average GPU memory usage.

**Figure 26 sensors-23-06138-f026:**
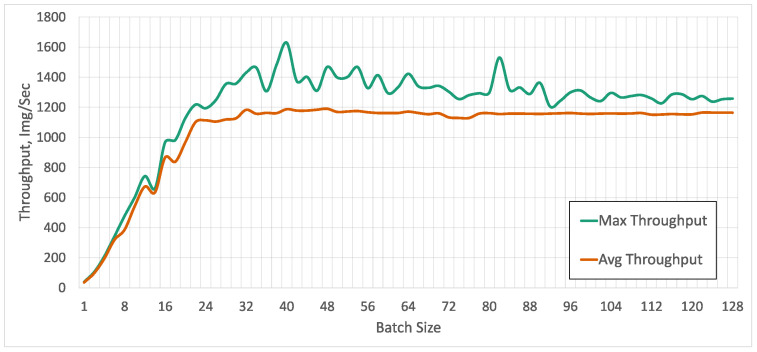
Maximum and average image throughput with a single model for different numbers of concurrently processed images.

**Figure 27 sensors-23-06138-f027:**
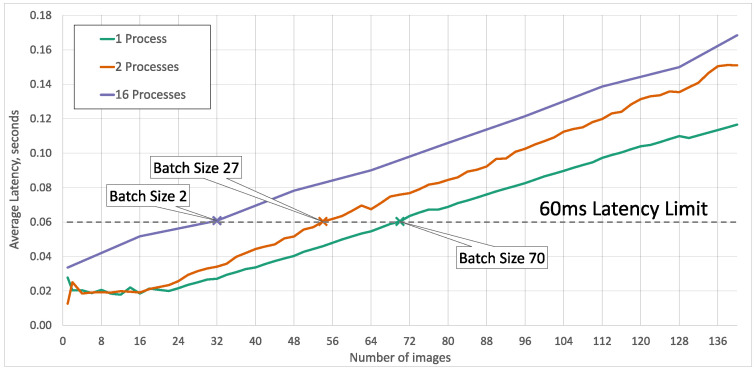
Processing latency for different numbers of processes, with a maximum possible batch size under a realistic latency limit of 60 ms.

**Table 1 sensors-23-06138-t001:** Comparison of file size, average latency and mean average precision for different models.

Model	Size	Latency	mAP@[0.5:0.95]
YOLOv5s	14.3 MB	19.82 ms	0.638
YOLOv5m	42.2 MB	25.00 ms	0.649
YOLOv5l	92.9 MB	32.87 ms	0.672
YOLOv5x	173.2 MB	38.77 ms	0.659
SSD320	96.9 MB	23.58 ms	0.114
SSD640	96.9 MB	23.73 ms	0.233
Faster-RCNN	162.0 MB	59.69 ms	0.460

**Table 2 sensors-23-06138-t002:** Performance of different models on Nvidia A100 at 16.66 Hz camera frame rate.

Running Model(-s)		Average Latency, ms	Waiting for Next Frame, ms
HDM only		26.5	31.6
ODM only		27.2	31.4
Concurrently	HDM	28.0	32.9
	ODM	28.4	32.2
CDM		28.9	29.9

**Table 3 sensors-23-06138-t003:** Processing latency (preprocessing, inference, NMS, and postprocessing) for the CDM on different GPUs.

GPU	Latency, ms		
	Min	Mean	Max
RTX 3080	20.6	25.7	39.8
A100	25.4	30.1	39.6
RTX 2080 Ti	35.7	40.4	55.1

**Table 4 sensors-23-06138-t004:** Latency (preprocessing, inference, NMS and postprocessing) on Nvidia Titan RTX using different CUDA versions.

Concurrent Images (Processes × Batch Size)	Average Latency, ms	
	CUDA 10.2	CUDA 11.1
1 (1 × 1)	12.7	19.1
10 (1 × 10)	33.4	36.1
10 (10 × 1)	59.2	57.1
16 (1 × 16)	50.5	56.6
16 (2 × 8)	50.3	53.2
16 (4 × 4)	50.2	54.4
16 (8 × 2)	61.5	60.2
16 (16 × 1)	93.4	-
19 (19 × 1)	59.4	65.3

## Data Availability

The dataset for object and hand detection presented in this study is available on request from the corresponding author.

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
