# Peer review of "Data Fusion for Cross-Domain Real-Time Object Detection on the Edge"

_sensors, 2023, doi:10.3390/s23136138_

Round 1

Reviewer 1 Report

The overall content of the article was informative. I believe there is an opportunity to provide a more detailed analysis of the relevance of the works in the related work section, explicitly concerning the problem you are trying to solve. Additionally, it would be beneficial to identify better the shortcomings of the current state of the art and how your research addresses those gaps.

To accomplish this, I suggest comprehensively analyzing the works you include in the related work section. Rather than just listing the references, aim to explain each work's specific contributions and findings and highlight their relevance to your research problem. Identify the aspects of the existing approaches that directly apply to your work and explain how your approach builds upon or differs from these current techniques (multi-task learning, ensemble learning, and sequential training). This analysis will help readers understand the context and significance of the referenced works and how they relate to your research.

Moreover, it would be valuable to clearly articulate the limitations or shortcomings of your field's current state of the art. Discuss the challenges or gaps the existing techniques face and how they might hinder the achievement of desired outcomes. Doing so can position your research as an advancement that addresses these limitations or provides novel solutions to overcome the identified challenges.

By providing a more detailed analysis of the relevance of the works in the related work section and identifying the shortcomings of the current state of the art, you will strengthen the foundation of your research and enhance the readers' understanding of your work's unique contributions to the field.

I believe that enhancing the related work section by incorporating additional references and explaining the existence of the most common approaches for the problem (multi-task learning, ensemble learning, and sequential training) would significantly enhance the article's value and relevance. In a quick search, I found several references in the literature to use these approaches in related problems.

Firstly, I recommend including a recent work titled "Multi-task Learning on the Edge for effective gender, age, ethnicity, and emotion recognition" by Pasquale Foggia, Antonio Greco, Alessia Saggese, and Mario Vento. This paper, published in Engineering Applications of Artificial Intelligence (Volume 118, 2023), discusses applying multi-task learning techniques for various recognition tasks on edge devices.

Secondly, I suggest including a paper titled "FedNets: Federated Learning on Edge Devices Using Ensembles of Pruned Deep Neural Networks" by B. Alhalabi, S. Basurra, and M. M. Gaber, published in IEEE Access. This paper, published in 2023, focuses on federated learning techniques using ensembles of pruned deep neural networks on edge devices.

Furthermore, I recommend including a recent paper titled "Real-time object detection using a domain-based transfer learning method for resource-constrained edge devices" by D. Kim, S. Lee, N.-M. Sung, and C. Choe. This paper, presented at the 2023 International Conference on Artificial Intelligence in Information and Communication (ICAIIC), explores a domain-based transfer learning approach for real-time object detection on resource-constrained edge devices.

Lastly, I would like to draw your attention to a paper titled "Finicky transfer learning—A method of pruning convolutional neural networks for cracks classification on edge devices" by Å»arski, M., Wójcik, B., Książek, K., and Miszczak, J. A. published in Computers-Aided Civil and Infrastructure Engineering. This paper, published in 2022, proposes a method of pruning convolutional neural networks for crack classification on edge devices, which aligns with the focus of your article. Including this reference would highlight an effective technique for addressing resource constraints and optimizing neural networks for edge devices.

By incorporating these additional references into the related work section of your article, you can provide readers with a broader understanding of the existing techniques and recent advancements in the field. Updating the references to include more recent works will also ensure that readers can access the latest and most relevant research in the domain.

You must explain why the Mean Average Precision (mAP) measurement with an Intersection over Union (IoU) threshold of 0.5, referred to as [email protected], is relevant for the evaluation in your study.

Section 5.2 in your article appears to be deviating from the primary focus and theme. To ensure coherence and maintain the flow of your research, I recommend considering two options:

Firstly, you could incorporate the synthetic and experimental evaluations on the same "level" within the article. This would involve reorganizing the text and content to integrate the findings from the merging of different splits of the COCO benchmark dataset with your initial object and gesture datasets. By doing so, you would present a comprehensive analysis of the impact of model fusion across various datasets and highlight the generalizability of your approach.

Alternatively, if the investigation of synthetic benchmarks and their impact on detection accuracy is not directly aligned with the main objectives of this article, you may consider removing Section 5.2 entirely. Instead, you could focus on addressing the generalization aspect in a separate, dedicated article that specifically explores the effects of merging different datasets and evaluating the impact on detection accuracy.

I have noticed that your paper includes several references to arXiv articles, and I wanted to provide some guidance on including arXiv references in scientific papers.

ArXiv is a platform for sharing preprints, which are early versions of research papers that have not undergone formal peer review. Including arXiv references can be valuable if you want to acknowledge and provide visibility to the preliminary work that influenced your research. However, it is essential to note that arXiv papers have not undergone the same level of scrutiny as peer-reviewed publications.

When deciding whether to include an arXiv reference, consider the availability of a peer-reviewed version of the same work. Suppose the arXiv paper has been subsequently published or accepted by a peer-reviewed venue. In that case, citing the final, peer-reviewed version rather than the arXiv preprint is generally preferable. This helps ensure the credibility and reliability of your references.

I hope you find these suggestions valuable and consider implementing them to enhance the quality and impact of your article. Thank you for your attention, and I look forward to seeing the improved version of your work.

Regarding the organization and content of the subsections in your paper, I recommend ensuring that all subsections have some text, whether it's a summary of the section's purpose, contents, organization, or objectives. This will help readers navigate your article more effectively and understand the context of each subsection. Including a summary or goal at the beginning of each subsection provides readers with a clear understanding of what to expect in that particular section. It sets the stage for the subsequent content and helps readers grasp your article's overall structure and flow. Consider explicitly stating each subsection's goals, aims, or prominent points, thereby guiding readers and establishing a solid narrative. Including such text in all subsections will ensure that each section contributes meaningfully to your article's overall coherence and readability. It will also help readers quickly navigate the paper and locate specific information of interest.

I would advise ensuring coherence by consistently using "Fig." or "Figure" throughout the article when referring to figures and using capital letters when identifying references to tables and figures to maintain clarity and consistency in your writing. Also, Figure 15 is not referenced in the text.

Author Response

We appreciate the reviewer's evaluation of our article and their recognition of its informative content. In response to their valuable feedback, we acknowledge the opportunity to enhance the related work section by providing a more comprehensive analysis of the relevance of prior studies specifically related to the problem we aim to address. Additionally, we recognize the importance of better identifying and discussing the shortcomings of the current state of the art, emphasizing how our research effectively bridges those gaps. In this rebuttal, we address these suggestions and present our revised manuscript accordingly. 

> Improve the "related work" section by explaining more clearly the contributions and shortcomings of different techniques, described in the references, and how our work improves upon them, add suggested references: 

>> "Multi-task Learning on the Edge for effective gender, age, ethnicity, and emotion recognition" by Pasquale Foggia, Antonio Greco, Alessia Saggese, and Mario Vento. 

We appreciate this suggestion. The referenced paper provides an excellent addition to our research. 

>> FedNets: Federated Learning on Edge Devices Using Ensembles of Pruned Deep Neural Networks" by B. Alhalabi, S. Basurra, and M. M. Gaber 

We acknowledge the potential benefits of incorporating federated learning in our research. However, considering the focus and scope of our paper, delving extensively into this topic may result in unnecessary complexity. To avoid this we decided not to include this reference. 

>> "Real-time object detection using a domain-based transfer learning method for resource-constrained edge devices" by D. Kim, S. Lee, N.-M. Sung, and C. Choe 

This is a very good reference, but very similar to "Towards automatic model specialization for edge video analytics" which we already have in our paper. 

>> "Finicky transfer learning—A method of pruning convolutional neural networks for cracks classification on edge devices" by Å»arski, M., Wójcik, B., Książek, K., and Miszczak, J. A. 

This is an excellent suggestion; we thank you again for this reference. It has been incorporated into our related work section. 

> Explain why we use [email protected]    

Thank you for this advice, we have added a paragraph to our article to explain the benchmarks used in our research. 

> Rewrite section 5.2, make it more relevant to the primary focus and theme of the paper 

Thank you for your valuable suggestion. We have taken it into consideration and made the necessary adjustments to address the coherence and flow of our research. Specifically, we have incorporated the detailed description of the synthetic dataset splits into Section 4 where we discuss our own dataset. Additionally, we have reorganized Section 5 to enhance the clarity of our performance and accuracy investigation results. 

> Reconsider the arXiv references, as they are not peer-reviewed (often preprints) 

We acknowledge the importance of emphasizing the credibility and reliability of our references. While we agree that citing the final, peer-reviewed versions is generally preferable, we considered the value and relevance of the arXiv articles we referenced. We have taken your suggestion into consideration and have made efforts replace most of our arXiv references with recently published versions, wherever possible, thus ensuring the most up-to-date and rigorously reviewed sources. 

> Fill empty space between section/chapter names, make "Fig.", "Figure" and "Table" more coherent, fix reference to Fig. 15 

We have taken immediate action to rectify this by ensuring that all subsections now contain concise summaries that provide an overview of the section's purpose, contents, organization, or objectives. 

Additionally, we have carefully addressed the issues regarding the consistency in referencing figures and fixed the missing reference to Figure 15. 

Reviewer 2 Report

The work in this paper is interesting! It shows some critical findings for cooperative perception. The feasibility of using a single object detection model (YOLOv5) with the benefit of reduced computational resources against the potentially more accurate independent and specialized models is tested. Using one single convolutional neural network (for object detection and hand gesture classification) instead of two separate ones can reduce resource usage by almost 50% has been found. This finding is of much importance for the multi-agent applications for indoor robots or outdoor connected automated vehicles. Well done. I hope this paper can be accepted although I have some quick minor questions:

1) Please highlight the contributions of this work explicitly in the introduction;

2) It will be helpful if the authors can add a system architecture diagram along with the physical figure 1.

3) Is that possible to add some description for the Yolo network in the paper at a very brief level? In this way, readers will be able to understand the paper better.

4) The experimental tests are quite sufficient to validate the findings and are organized well.

5) I believe the work in this paper can be applied to other areas such as connected automated vehicles and unmanned aerial vehicles. So, I hope the authors can include papers in these fields to discuss the potential of the findings in this work. For the connected automated vehicles, please include the paper: an automated driving systems data acquisition and analytics platform. Having them in the paper will increase the interest in this work in wider communities.

Author Response

We appreciate your positive feedback and acknowledgement of the significance of our work in exploring cooperative perception. We sincerely thank you for your encouraging remarks and look forward to addressing your minor questions in detail to further improve the paper. 

> Highlight the contributions of the work more explicitly in the introduction 

Thank you for your comment. We have taken your feedback into consideration and expanded the introduction section accordingly to better emphasize the specific contributions of our research. 

> Add a system architecture diagram along with the physical figure 1  

We appreciate the suggestion, however we want to inform you that we have already provided a comprehensive system pipeline representation in Figure 2. This figure presents a schematic diagram of the visual inspection pipeline, showcasing the key components and their interactions within the system. 

> Add short description of the YOLOv5 architecture 

We have taken your feedback into consideration and have now added a concise description of the YOLO model architecture in the article. 

> Suggest to apply the findings to other fields, i.e. automated vehicles, add a reference: 

>> "an automated driving systems data acquisition and analytics platform" 

We appreciate your recommendation to include a paper on an automated driving systems data acquisition and analytics platform, as it aligns with the potential of our findings in the field of connected automated vehicles. We agree that our work can have broader applications beyond the scope of cooperative perception and have included the suggested reference in our paper 

Reviewer 3 Report

The paper examines various aspects of automated object recognition technologies, with a particular focus on static gestures. The authors conduct experiments to address memory bottlenecks in the GPU and enable parallel execution of multiple object recognition tasks. The research results are evaluated through testing and comparative analysis using a custom dataset that includes both objects in production and static gestures. The comparative analysis shows that the performance quality is still affected by the input data, which can be influenced by the choice of datasets or visual corpus. The article includes helpful explanatory figures and tables, as well as technical descriptions relevant to the chosen journal. However, the following shortcomings of the paper need to be addressed.

Shortcomings:

1) Firstly, the paper lacks a comprehensive description of the factors that may influence the recognition of static and dynamic gestures. The authors briefly mention the potential use of gestures for non-verbal control and interaction with robotic platforms or collaborative robots. It would be beneficial to expand this section to clarify that, despite the significant practical potential, the problem of effectively recognising any gesture remains unsolved due to significant variations in their semantic-syntactic structures. Consequently, unambiguous translation of not only control gestures but also gesture languages into textual representations is currently not feasible. As a result, fully automated models and methods for recognising multiple static and dynamic gestures do not currently exist. Creating such comprehensive models would require deep semantic analysis, which is currently limited due to deficiencies in text analysis algorithms, knowledge bases, visual corpora, and other related factors. In addition, it is worth noting that sign language recognition, beyond control gestures, is a current and relevant issue, particularly for people with hearing impairments. This further emphasizes the importance of the chosen scientific research, as gesture recognition falls within the domain of assistive technologies. Furthermore, it is crucial to recognize that the operation of robots in manufacturing and other contexts is not limited to people without hearing impairments; the deaf community, which relies on gestures as a familiar mode of communication, can also freely interact with intelligent systems installed on robotic platforms, highlighting the importance of this research in today's world.

2) The authors of this article have built their visual corpus from objects and static gestures. Therefore, it is also reasonable to explain that at the moment there are many problems that arise due to the lack of universal techniques for creating not only unimodal, but also multimodal corpus with different objects, providing increased efficiency of machine learning and accuracy of automatic recognition of objects using different video capture devices, which provide not only high-quality images in optical mode, but also additional data about the coordinates of graphic areas. Indeed, it is possible to create visual cases based on such universal techniques, which can then be used, among other things, for the analysis of objects of interest.

3) It is useful to extend the description of related work (section 2) with modern and, importantly, better methods of gesture recognition. Modern methods can be found on paperwithcode, on the SOTA page of a particular gesture corpus. If we look at the modern and well-known AUTSL corpus (https://paperswithcode.com/sota/sign-language-recognition-on-autsl), the best methods are STF+LSTM (1st place), SAM-SLR (RGB-D), Ensemble - NTIS, MViT-SLR, FE+LSTM. All this will show that the authors are aware of the best results achieved on large enclosures (AUTSL and others). Otherwise, what happens now is that the authors mention their assembled hull with static gestures and other objects, but do not describe other best practices. In SOTA, AUTSL methods build on Mediapipe and other approaches and describe their applications on mobile and robotic devices (STF+LSTM), which is also relevant for robot control. This will also be useful information for future readers of the paper.

4) There are no references to related works from the world scientific community (2021-23), which are constantly presented at video modality-oriented conferences (CVPR, ICCV, INTERSPEECH, ICASSP, EUSIPCO, ICMI, SPECOM and others) or in first quartile journals (Sensors, Neurocomputing and others). However, if the authors correct shortcomings 2-3, this observation will resolve itself.

5) Why was YOLOv5 chosen instead of newer versions?

6) More details are needed about the model learning experiments. For example, it is important to explain why the attention module was not used, why modern MixUp techniques were not used, and whether techniques such as cosine annealing were used.

7) In addition, it is crucial to provide more information about future work to improve the comprehensiveness of the article.

8) The abstract should be expanded to provide a more comprehensive overview of the research.

9) The keywords should also be expanded to include a wider range of relevant terms.

10) Finally, the article requires minor revisions to address spelling and punctuation errors, which will improve its overall style and readability.

Incorporating the suggested additions and revisions will undoubtedly improve the quality and completeness of the research.

Moderate editing of English language required.

Author Response

We appreciate the reviewer's comprehensive assessment of our paper, and we are grateful for the reviewer's acknowledgement of the informative figures, tables, and technical descriptions that align with the standards of the chosen journal. In our rebuttal, we will address the identified shortcomings and provide further clarification and improvements to ensure the strength of our research. 

> Talk more about the problems of recognizing static and dynamic gestures, particularly for sign-language 

Thank you for your feedback. While we acknowledge the importance of creating universal techniques for multimodal visual corpora and the potential benefits they offer for object recognition, our research primarily focuses on the technical aspects of edge computing and convolutional neural networks. Therefore, we would like to avoid delving too deep into that specific topic, since it is not directly relevant to the context and scope of our paper. 

> About different approaches for generating a visual corpus from objects and static gestures 

Thank you for highlighting the importance of universal techniques for creating multimodal corpora with various objects to enhance machine learning efficiency and object recognition accuracy. However, our paper primarily focuses on technical aspects related to edge computing and CNN, which makes this particular aspect less relevant within the context of our research. 

> About RGB-D and LSTM based sign language recognition 

We appreciate the relevance of these approaches, however our research primarily focuses on static gestures and does not involve dynamic gestures or the use of RGB-D cameras. Therefore, the mentioned approaches utilizing STF+LSTM and other methods based on Mediapipe are not directly applicable to the scope of our work. Based on our current use cases and applications we needed to use RGB cameras which motivated our decision to use this modality. 

> More references to related works from video modality-oriented conferences and first quartile journals 

While we understand the importance of such references, it is crucial to note that our paper's scope is primarily focused on the application of RGB modality for gesture recognition within our specific context. Our decision to utilize RGB modality was driven by the requirements of our application, and we firmly believe it is sufficient for our objectives.  

We will consider your feedback as we plan for future research endeavours, where we can delve into the topic of different modalities for object detection in a more comprehensive manner. 

> Why was YOLOv5 chosen instead of newer versions? 

We acknowledge the reviewer's comment regarding the choice of YOLOv5 instead of newer versions. At the time of our research, YOLOv5 was the latest available version. We have since experimented with YOLOv8 and found that while it may offer improved accuracy, there is no significant difference in terms of processing speed, at least for our application. More importantly, we have also observed that the core concept of improved accuracy of a combined model when using data fusion remains applicable (see attached Figure for the comparisons on YOLOv8m).  

[figure here]

We will provide additional results for the newer versions of the YOLO model in our future work. 

> More details are needed about the model learning experiments 

In contrast to existing approaches that primarily concentrate on model modification, our research adopts a methodology that does not necessitate model alterations. By employing this approach, our findings possess the potential for broad generalization across diverse models, both present and future. 

> Provide more information about future work, the abstract and keywords should be expanded 

Thank you for your valuable feedback. We have taken your suggestions into consideration and made the necessary revisions to address the issues you raised. Specifically, we have added a section dedicated to discussing future work, expanded the abstract to provide a more comprehensive overview of our research, and included a wider range of relevant keywords. We appreciate your input, as these improvements enhance the overall quality and completeness of our article. 

> Spelling and punctuation errors 

Thank you for pointing this out. We have thoroughly proofread our paper once again and made the necessary corrections to rectify any identified errors. 

Round 2

Reviewer 3 Report

The authors have skilfully expanded on the concepts presented in the article, going into greater detail and answering questions that have arisen. The resulting work not only appears comprehensive, but also exhibits a level of completeness that warrants its strong recommendation for publication. The authors' thoroughness in providing comprehensive explanations and responses contributes greatly to the overall quality of the article, making it an excellent candidate for further consideration in the publication process.

In my previous review, I suggested the inclusion of dynamic gesture recognition methods. While I understand that the authors have chosen not to experiment with dynamic gestures in their article, I still believe that mentioning the existence of such gestures and current methods for recognising them would enhance the overall value of the article. This addition would provide readers with a broader understanding of the topic and its potential applications. However, I acknowledge that the authors may have valid reasons for excluding this information, and I respect their decision. Therefore, I reiterate that this is only a recommendation, and I leave it to the authors to include or exclude it as they see fit.

Minor editing of English language required.